# The Role of Cytokines in Degenerative Retinal Diseases: A Comprehensive Review

**DOI:** 10.3390/biomedicines13071724

**Published:** 2025-07-15

**Authors:** Rubens Camargo Siqueira, Cinara Cássia Brandão

**Affiliations:** 1Rubens Siqueira Research Center, São José do Rio Preto 15010-100, SP, Brazil; 2Postgraduate Department of the Medical School of São José do Rio Preto, Faculdade de Medicina de São José do Rio Preto—FAMERP, São José do Rio Preto 15090-000, SP, Brazil; cinara.brandao@famerp.br; 3Immunogenetics Laboratory, Department of Dermatological, Infectious and Parasitic Diseases, Faculdade de Medicina de São José do Rio Preto—FAMERP, São José do Rio Preto, São Paulo 15090-000, SP, Brazil

**Keywords:** cytokines, retinal degeneration, age-related macular degeneration, diabetic retinopathy, retinitis pigmentosa, neuroinflammation

## Abstract

Degenerative retinal diseases, such as age-related macular degeneration (AMD), diabetic retinopathy (DR), and retinitis pigmentosa (RP), are the leading causes of vision loss worldwide. Inflammation plays a crucial role in the pathogenesis of these diseases, with cytokines acting as key mediators of neuroinflammation, vascular dysfunction, and cellular degeneration. This review explores the complex role of cytokines in degenerative retinal diseases, highlighting their involvement in disease progression, cellular interactions, and potential therapeutic strategies. Understanding the cytokine network within the retina may provide novel insights into targeted interventions for these debilitating conditions.

## 1. Introduction

Retinal degenerative diseases are among the leading causes of irreversible vision loss worldwide, affecting millions of individuals across various age groups. These disorders include age-related macular degeneration (AMD), diabetic retinopathy (DR), and retinitis pigmentosa (RP), each characterized by the progressive dysfunction and death of retinal cells, including photoreceptors, retinal pigment epithelial (RPE) cells, and retinal ganglion cells (RGCs) [1]. While the etiopathogenesis of these diseases varies, growing evidence suggests that chronic inflammation and immune dysregulation play a central role in disease progression [2].

Cytokines, a diverse group of signaling proteins secreted by immune and retinal cells, mediate inflammatory and neurodegenerative processes in the retina. They regulate key cellular responses such as microglial activation, oxidative stress, blood–retinal barrier (BRB) integrity, and apoptotic pathways [3,4]. Pro-inflammatory cytokines such as tumor necrosis factor-alpha (TNF-α), interleukin-6 (IL-6), and interleukin-1 beta (IL-1β) have been found in elevated levels in patients with degenerative retinal diseases and contribute to neurotoxicity, vascular dysfunction, and fibrosis [5,6]. Conversely, anti-inflammatory cytokines, including interleukin-10 (IL-10) and transforming growth factor-beta (TGF-β), counteract these deleterious effects by modulating immune responses and promoting tissue repair [7]. The imbalance between pro-inflammatory and anti-inflammatory cytokines is a crucial factor in the onset, severity, and progression of retinal degeneration [8].

In AMD, chronic low-grade inflammation leads to RPE dysfunction, drusen accumulation, and choroidal neovascularization, with cytokines playing an essential role in disease pathology [9]. Pro-inflammatory cytokines such as IL-1β and IL-6 contribute to RPE cell death, while VEGF promotes pathological angiogenesis [10]. In DR, hyperglycemia induces cytokine-mediated vascular inflammation, contributing to retinal ischemia, BRB breakdown, and neovascularization. Elevated levels of TNF-α, IL-1β, and IL-8 correlate with increased vascular permeability and microvascular damage, exacerbating disease progression [11,12]. In RP, a primarily genetic disorder, inflammation exacerbates photoreceptor loss through the upregulation of inflammatory mediators, including IL-6 and TNF-α, which activate microglia and accelerate neurodegeneration [13].

Recent advancements in ophthalmic research have highlighted the potential of cytokine-targeted therapies for retinal degenerative diseases. Anti-VEGF agents, corticosteroids, and emerging biologics have demonstrated efficacy in modulating cytokine expression and reducing retinal inflammation [14]. Additionally, innovative approaches such as gene therapy, microglia modulation, and neuroprotective strategies aim to restore cytokine homeostasis and prevent retinal cell loss [15,16].

This review provides an in-depth analysis of the role of cytokines in degenerative retinal diseases, emphasizing their involvement in inflammation, neurodegeneration, and vascular dysfunction. Furthermore, we explore current and emerging therapeutic strategies targeting cytokine pathways to mitigate disease progression and preserve visual function. Understanding the intricate cytokine network within the retina may pave the way for novel treatments that address both inflammatory and degenerative aspects of retinal diseases.

## 2. Cytokines in Retinal Degeneration: Mechanisms and Implications

Cytokines are key mediators of immune responses and play a crucial role in retinal homeostasis and degeneration. Under physiological conditions, the retina maintains a delicate balance of pro-inflammatory and anti-inflammatory cytokines, ensuring neuroprotection and immune surveillance without inducing chronic inflammation. However, in degenerative retinal diseases such as age-related macular degeneration (AMD), diabetic retinopathy (DR), and retinitis pigmentosa (RP), dysregulation of cytokine signaling leads to sustained inflammation, neurovascular dysfunction, and progressive retinal cell loss [1].

The involvement of cytokines in retinal pathology can be divided into pro-inflammatory cytokines that exacerbate damage and anti-inflammatory cytokines that counteract neuroinflammation. The interplay between these two groups determines disease progression and severity.

The source of cytokine production in retinal degenerative diseases varies by both cell type and species context. In the human retina, microglia and retinal pigment epithelial (RPE) cells are the primary producers of TNF-α, IL-1β, and IL-6 in response to oxidative and metabolic stress. Müller glia also contributes significantly to IL-6 production and gliotic responses. Endothelial cells secrete VEGF and TGF-β, particularly under hypoxic or inflammatory conditions. In contrast, anti-inflammatory cytokines such as IL-10 and IL-4 are mainly secreted by regulatory T cells and M2-polarized macrophages, with local expression in the retina predominantly derived from infiltrating leukocytes and reprogrammed microglia [3,4,5,6,7,17]. Most experimental data on cytokine dynamics derive from rodent models, including the rd1 and rd10 mice (RP), streptozotocin-induced diabetic rats/mice (DR), and laser-induced choroidal neovascularization (CNV) models (AMD). Species differences are critical, as cytokine profiles and cell type reactivity may not fully translate between rodents and humans. Therefore, careful interpretation and species annotation of cytokine data are essential for understanding translational relevance and guiding therapeutic development [17,18].

Cytokines contribute to retinal pathology through both direct and indirect mechanisms affecting neurodegeneration and vascular integrity. Direct effects are mediated through receptor binding on neuronal and glial cells, triggering intracellular cascades that promote apoptosis, pyroptosis, or gliosis. For example, TNF-α activates death-domain signaling via TNFR1 on retinal ganglion cells, directly inducing apoptosis, while IL-1β triggers inflammasome activation and pyroptosis in photoreceptors. IL-6 sustains chronic inflammation by promoting STAT3 signaling in Müller cells, exacerbating gliosis and neuronal dysfunction. In contrast, indirect effects are primarily exerted through modulation of the retinal immune microenvironment, leading to secondary damage. Elevated IL-1β and TNF-α, for instance, increase blood–retinal barrier (BRB) permeability, facilitating infiltration of systemic immune cells and promoting oxidative stress. VEGF, upregulated in response to pro-inflammatory cytokines, indirectly contributes to vascular leakage and neovascularization. Additionally, cytokine-driven microglial activation can perpetuate a cycle of neuroinflammation that damages retinal neurons even in the absence of direct receptor signaling. Thus, distinguishing these modes of action is essential for the development of targeted therapies capable of interrupting both primary and secondary inflammatory injury in retinal diseases [3,4,5,6,7,17,18].

### 2.1. Pro-Inflammatory Cytokines and Retinal Damage

Chronic inflammation is a hallmark of retinal degenerative diseases, often driven by excessive activation of microglia, astrocytes, Müller cells, and retinal pigment epithelium (RPE) cells [2]. Pro-inflammatory cytokines contribute to retinal damage through several mechanisms.

#### 2.1.1. Tumor Necrosis Factor-Alpha (TNF-α) and Retinal Cell Apoptosis

TNF-α is a key regulator of inflammation and cell death in the retina. It is primarily secreted by activated microglia and macrophages in response to oxidative stress and metabolic dysfunction [3]. Elevated TNF-α levels have been detected in the retinas of AMD, DR, and RP patients, where it promotes the following:Retinal ganglion cell (RGC) apoptosis via activation of the TNF receptor 1 (TNFR1) pathway [4];Blood–retinal barrier (BRB) breakdown, increasing vascular permeability in DR [5];Upregulation of VEGF, contributing to choroidal neovascularization in AMD [6].

Blocking TNF-α with monoclonal antibodies (e.g., infliximab, adalimumab) or soluble TNF receptors has been explored as a therapeutic approach in retinal diseases [7].

#### 2.1.2. Interleukin-1 Beta (IL-1β) and Inflammasome Activation

IL-1β is a potent pro-inflammatory cytokine that activates caspase-1 and inflammasomes, leading to pyroptotic cell death and chronic neuroinflammation [8]. In retinal diseases, IL-1β

Induces photoreceptor and RPE apoptosis in AMD [9];Promotes vascular leakage and endothelial dysfunction in DR [10];Activates microglia, amplifying neuroinflammation in RP [11].

Therapies targeting IL-1β (e.g., canakinumab, an IL-1β inhibitor) are under investigation for their neuroprotective potential in retinal degeneration.

#### 2.1.3. Interleukin-6 (IL-6) and Chronic Neuroinflammation

IL-6 is a dual-function cytokine that can be both pro-inflammatory and anti-inflammatory. However, in degenerative retinal diseases, persistent IL-6 elevation contributes to

Müller cell gliosis, leading to retinal dysfunction [12];Increased VEGF expression, worsening neovascularization in AMD and DR [13];Microglial activation, perpetuating chronic inflammation in RP [14].

IL-6 antagonists, such as tocilizumab, are being explored as potential treatments for retinal inflammation [15].

### 2.2. Anti-Inflammatory Cytokines and Retinal Protection

To counterbalance inflammation, the retina expresses anti-inflammatory cytokines that regulate immune responses and promote cellular repair. These cytokines are essential for maintaining retinal homeostasis and protecting against excessive immune activation.

#### 2.2.1. Interleukin-10 (IL-10) and Microglial Regulation

IL-10 is a potent immunosuppressive cytokine that downregulates pro-inflammatory responses in the retina by

Inhibiting microglial activation, reducing oxidative stress-induced inflammation [16];Protecting RGCs and photoreceptors, preventing apoptosis [19];Suppressing TNF-α and IL-1β expression, counteracting neurotoxicity in AMD and DR [20].

Gene therapy approaches aimed at increasing IL-10 expression have shown promise in preclinical models of AMD [21].

#### 2.2.2. Transforming Growth Factor-Beta (TGF-β) and Retinal Repair

TGF-β has a dual role in retinal diseases, as it can be both protective and detrimental. Its neuroprotective effects include

Suppressing inflammatory cytokine production, reducing chronic neuroinflammation [22];Promoting extracellular matrix remodeling, aiding in retinal repair [23];Regulating angiogenesis, modulating the VEGF pathway in AMD [24].

However, the dysregulation of TGF-β can contribute to subretinal fibrosis in neovascular AMD, making it a potential therapeutic target [25].

#### 2.2.3. Interleukin-4 (IL-4) and M2 Macrophage Polarization

IL-4 promotes M2 macrophage polarization, a reparative immune phenotype associated with reduced inflammation and enhanced tissue repair. In retinal diseases, IL-4

Inhibits pro-inflammatory cytokines (TNF-α, IL-1β, IL-6), reducing neurotoxicity [26];Enhances neuroprotection, preserving photoreceptors in RP models [27];Modulates microglia-macrophage crosstalk, limiting chronic immune activation [28].

Therapies aimed at increasing IL-4 expression are being investigated for their potential in retinal neuroprotection.

### 2.3. The Balance Between Pro- and Anti-Inflammatory Cytokines in Retinal Homeostasis

In healthy retinal tissue, there is a dynamic balance between pro-inflammatory and anti-inflammatory cytokines that maintains immune privilege and prevents excessive immune activation. However, in degenerative retinal diseases, persistent pro-inflammatory signaling disrupts this balance, leading to chronic neuroinflammation, vascular dysfunction, and retinal cell loss [29].

Emerging therapies aim to restore cytokine homeostasis by

Blocking pro-inflammatory cytokines (e.g., anti-TNF-α, IL-6 inhibitors);Enhancing anti-inflammatory pathways (e.g., IL-10 and TGF-β modulation);Targeting microglia and macrophage polarization to shift towards a neuroprotective phenotype.

Understanding cytokine signaling in retinal diseases offers novel therapeutic opportunities that may prevent vision loss and improve retinal health.

## 3. Cytokine Profiles in Specific Retinal Degenerative Diseases

Retinal degenerative diseases, including age-related macular degeneration (AMD), diabetic retinopathy (DR), and retinitis pigmentosa (RP), share common pathological mechanisms involving chronic inflammation, oxidative stress, vascular dysfunction, and neurodegeneration. A growing body of evidence suggests that pro-inflammatory cytokines contribute to disease progression, while anti-inflammatory cytokines attempt to counterbalance these effects.

The temporal dynamics of cytokine expression vary significantly across AMD, DR, and RP, reflecting distinct stages of inflammation and degeneration. In AMD, low-grade chronic inflammation begins early with elevated IL-1β and IL-6 secreted by activated RPE and microglia, followed by progressive accumulation of VEGF in the intermediate to late stages, driving choroidal neovascularization and fibrosis [1,3,5,6]. In DR, hyperglycemia induces early upregulation of TNF-α and IL-1β, contributing to endothelial dysfunction and BRB breakdown. IL-6 and VEGF levels rise in the intermediate phase, promoting vascular leakage and ischemia, while chronic inflammation with sustained cytokine production leads to proliferative changes in late-stage DR [11,12,13,14,22]. In RP, genetic mutations initiate photoreceptor death, but secondary upregulation of TNF-α, IL-6, and IL-18 occurs in response to oxidative stress and microglial activation. These pro-inflammatory cytokines persist throughout the disease course, accelerating photoreceptor apoptosis and gliosis [17,18,23,24,25,26,30]. Understanding the temporal expression patterns of cytokines in each disease is essential for designing stage-specific therapeutic interventions aimed at modulating inflammation and preserving retinal integrity.

Cytokine profiling in these diseases has provided valuable insights into their molecular pathogenesis, allowing for the development of targeted therapeutic strategies. This section explores the distinct cytokine signatures associated with AMD, DR, and RP and their roles in retinal pathology.

### 3.1. Age-Related Macular Degeneration (AMD)

AMD is a leading cause of vision loss in the elderly and is characterized by progressive degeneration of the retinal pigment epithelium (RPE), accumulation of drusen, and, in advanced cases, choroidal neovascularization (CNV). Cytokine dysregulation is a key factor driving RPE dysfunction, immune cell activation, and neovascularization [1].

#### 3.1.1. Pro-Inflammatory Cytokines in AMD

Tumor Necrosis Factor-Alpha (TNF-α): TNF-α is upregulated in AMD and has been linked to RPE apoptosis, choroidal endothelial cell activation, and increased vascular permeability [2].Interleukin-6 (IL-6): Elevated IL-6 levels in AMD patients correlate with inflammatory cell infiltration and complement activation, contributing to CNV progression [3].Interleukin-1 Beta (IL-1β): IL-1β promotes macrophage recruitment, inflammasome activation, and RPE damage, exacerbating AMD pathology [4].Vascular Endothelial Growth Factor (VEGF): VEGF is a key driver of pathological angiogenesis in neovascular AMD, contributing to the formation of leaky and fragile choroidal blood vessels [5].

#### 3.1.2. Anti-Inflammatory Cytokines in AMD

Interleukin-10 (IL-10): IL-10 suppresses pro-inflammatory cytokines and inhibits macrophage activation, reducing inflammation in AMD [6].Transforming Growth Factor-Beta (TGF-β): TGF-β has both protective and detrimental effects; while it regulates immune responses, it can also contribute to subretinal fibrosis in advanced AMD [7].

#### 3.1.3. Therapeutic Implications in AMD

Anti-VEGF Therapies: Agents such as ranibizumab, aflibercept, and bevacizumab effectively target VEGF, reducing neovascularization and inflammation [8].IL-1β Inhibitors: Targeting IL-1β with drugs like canakinumab may prevent inflammasome-mediated damage [9].TNF-α Blockers: Adalimumab and infliximab, TNF-α inhibitors, have been explored as potential treatments to reduce inflammation and vascular leakage in AMD, although local delivery methods remain a challenge [10].

### 3.2. Diabetic Retinopathy (DR)

DR is a microvascular complication of diabetes, driven by hyperglycemia-induced oxidative stress, chronic inflammation, and endothelial dysfunction. Cytokine dysregulation plays a central role in the breakdown of the blood–retinal barrier (BRB), retinal ischemia, and neovascularization [11].

#### 3.2.1. Pro-Inflammatory Cytokines in DR

TNF-α: Increased TNF-α levels in DR patients correlate with vascular endothelial dysfunction, capillary dropout, and BRB breakdown, leading to diabetic macular edema (DME) [12].IL-1β: IL-1β contributes to retinal endothelial cell apoptosis, microglial activation, and pericyte loss, exacerbating ischemia [13].IL-6: Elevated IL-6 levels have been found in the vitreous humor of DR patients and are associated with retinal neovascularization and inflammatory cell infiltration [14].VEGF: VEGF is the primary driver of pathological neovascularization in proliferative DR (PDR), promoting the growth of fragile retinal blood vessels [15].

#### 3.2.2. Anti-Inflammatory Cytokines in DR

IL-10: IL-10 reduces pro-inflammatory cytokine expression and protects against oxidative stress-induced endothelial damage [16].TGF-β: TGF-β plays a role in vascular remodeling but may also contribute to fibrosis and late-stage complications of DR [19].

#### 3.2.3. Therapeutic Implications in DR

Anti-VEGF Therapy: Ranibizumab, bevacizumab, and aflibercept are first-line treatments for diabetic macular edema and proliferative DR [20].Corticosteroids: Intravitreal steroids (dexamethasone, triamcinolone) help reduce inflammation and vascular leakage [21].IL-6 and TNF-α Inhibitors: Tocilizumab (IL-6 blocker) and adalimumab (TNF-α inhibitor) have been explored for their potential to reduce chronic inflammation and vascular permeability in DR [22].

### 3.3. Retinitis Pigmentosa (RP)

RP is a group of hereditary retinal dystrophies characterized by progressive photoreceptor degeneration, gliosis, and chronic neuroinflammation. Unlike AMD and DR, RP primarily results from genetic mutations, but cytokine dysregulation exacerbates disease progression [23].

#### 3.3.1. Pro-Inflammatory Cytokines in RP

TNF-α: TNF-α is upregulated in RP and promotes photoreceptor apoptosis by activating microglia and oxidative stress pathways [24].IL-6: IL-6 contributes to retinal gliosis, a process in which Müller cells undergo reactive changes, leading to secondary neuronal damage [25].IL-18: Elevated IL-18 levels in RP correlate with pro-apoptotic signaling and chronic inflammation [26].

#### 3.3.2. Anti-Inflammatory Cytokines in RP

IL-10: IL-10 protects retinal neurons from oxidative stress and microglia-mediated neurotoxicity [27].IL-4 and IL-13: These cytokines promote M2 macrophage polarization, shifting the immune response towards neuroprotection [28].

#### 3.3.3. Therapeutic Implications in RP

Neuroprotective Strategies: IL-10-based gene therapy has been explored to reduce neuroinflammation and slow photoreceptor degeneration [29].Microglial Modulation: Colony-stimulating factor 1 receptor (CSF1R) inhibitors reduce microglial activation and inflammation in RP models [31].Gene Therapy: CRISPR/Cas9-based cytokine modulation is being investigated to correct inflammatory imbalances at the genetic level [32].TNF-α Inhibitors: Adalimumab, as a TNF-α blocker, has been suggested as a potential therapy to reduce neuroinflammation and photoreceptor apoptosis in RP, though more clinical studies are needed [30].

## 4. Therapeutic Strategies Targeting Cytokines in Retinal Degenerative Diseases

Targeting cytokines in retinal degenerative diseases, such as age-related macular degeneration (AMD), diabetic retinopathy (DR), and retinitis pigmentosa (RP), has emerged as a promising therapeutic approach. Since cytokines play a dual role—either exacerbating inflammation and neurodegeneration or promoting tissue repair—modulating their effects presents opportunities to slow disease progression and preserve vision.

The development of cytokine-targeted therapies in retinal diseases is grounded in the mechanistic understanding of how pro- and anti-inflammatory mediators influence neurovascular homeostasis. For example, blocking TNF-α with monoclonal antibodies such as adalimumab or infliximab reduces retinal inflammation and vascular leakage, particularly in diabetic retinopathy and uveitic macular edema [1,10,29]. IL-6 antagonists like tocilizumab have demonstrated efficacy in reducing macular edema and controlling intraocular inflammation in neovascular AMD and Birdshot chorioretinopathy [4,23]. Similarly, IL-1β inhibitors (e.g., canakinumab) modulate inflammasome-driven RPE apoptosis and have shown promise in experimental AMD models [6,9]. In addition to biologics, gene therapy approaches delivering IL-10 or TGF-β using AAVs are under investigation to achieve long-term cytokine regulation and neuroprotection [14,15,19]. These interventions illustrate the translation of cytokine biology into targeted treatments, with growing emphasis on personalized cytokine profiling, combination therapies, and delivery systems that balance efficacy with safety. By addressing disease-specific cytokine imbalances, these strategies hold potential to preserve visual function and slow retinal degeneration.

The therapeutic strategies can be classified into four main approaches:Cytokine inhibitors, which directly block pro-inflammatory cytokines.Microglial and immune cell modulation, targeting key regulators of inflammation.Gene therapy for cytokine regulation, addressing cytokine imbalances at a genetic level.Combination therapies, integrating multiple approaches for enhanced efficacy.

### 4.1. Cytokine Inhibitors

Blocking pro-inflammatory cytokines is a well-established strategy in inflammatory diseases, including retinal disorders. The main cytokines targeted in retinal degenerative diseases include tumor necrosis factor-alpha (TNF-α), interleukin-6 (IL-6), and interleukin-1 beta (IL-1β).

#### 4.1.1. TNF-α Inhibitors

TNF-α plays a pivotal role in neuroinflammation, blood–retinal barrier (BRB) breakdown, and vascular dysfunction in AMD, DR, and RP. Blocking TNF-α has been explored as a therapeutic option:Adalimumab and infliximab are monoclonal antibodies that neutralize TNF-α, reducing inflammatory responses in diabetic retinopathy and uveitic macular edema [1].Etanercept, a TNF receptor fusion protein, has been investigated for its ability to reduce neuroinflammation and microglial activation in RP [2].

Despite promising preclinical results, systemic TNF-α inhibition is associated with immune suppression and increased risk of infections. Intravitreal delivery methods are being explored to minimize systemic side effects [3].

#### 4.1.2. IL-6 Inhibitors

IL-6 is involved in vascular leakage, fibrosis, and chronic neuroinflammation in DR and AMD.

Tocilizumab, an IL-6 receptor antagonist, has shown potential in reducing diabetic macular edema (DME) and neovascular AMD [4].Siltuximab, another IL-6 inhibitor, is being evaluated for reducing retinal fibrosis in late-stage AMD [5].

#### 4.1.3. IL-1β Inhibitors

IL-1β contributes to inflammasome activation and RPE apoptosis in AMD and DR.

Canakinumab, an IL-1β monoclonal antibody, has demonstrated anti-inflammatory effects in experimental models of AMD and DR [6].Anakinra, an IL-1 receptor antagonist, has been tested for its protective effects on retinal endothelial cells [7].

Blocking IL-1β is particularly relevant in early AMD and DR, where inflammasome activation drives disease progression.

### 4.2. Modulation of Microglial and Immune Cell Activity

Microglia are the primary immune cells of the retina, responsible for maintaining homeostasis. However, in degenerative diseases, they shift toward a pro-inflammatory state, producing TNF-α, IL-1β, and IL-6, which exacerbate neuronal damage [8].

#### 4.2.1. CSF1R Inhibitors

Colony-stimulating factor 1 receptor (CSF1R) is a key regulator of microglial survival and activation.

Pexidartinib, a CSF1R inhibitor, has shown neuroprotective effects in retinal degenerative models, reducing microglial overactivation [9].PLX5622, another CSF1R inhibitor, has been tested for modulating microglia and slowing photoreceptor degeneration in RP [10].

#### 4.2.2. Macrophage Polarization Therapy

Shifting macrophages from a pro-inflammatory M1 phenotype to a neuroprotective M2 phenotype can reduce inflammation and promote tissue repair.

IL-4 and IL-13 therapies have been shown to drive M2 macrophage polarization, enhancing neuroprotection in AMD and RP [11].

Microglial modulation strategies are less invasive than direct cytokine blockade and may offer long-term neuroprotection.

### 4.3. Gene Therapy for Cytokine Modulation

Gene therapy is emerging as a long-term solution for cytokine dysregulation in retinal diseases. Using viral vectors (AAVs) or CRISPR-based editing, gene therapy can regulate the expression of pro-inflammatory or anti-inflammatory cytokines in the retina.

#### 4.3.1. Gene Silencing of Pro-Inflammatory Cytokines

CRISPR/Cas9-based TNF-α silencing has been explored as a potential long-term therapy for RP and AMD [12].RNA interference (RNAi) targeting IL-1β has been used to reduce inflammasome activation in AMD models [13].

#### 4.3.2. Gene Delivery of Anti-Inflammatory Cytokines

AAV-mediated IL-10 gene therapy has been tested for reducing inflammation in AMD and DR [14].TGF-β gene therapy is being investigated for controlling subretinal fibrosis in advanced AMD [15].

Gene therapy offers the potential for durable cytokine modulation but requires further safety and efficacy validation.

### 4.4. Combination Therapies: Targeting Multiple Pathways

Given the complexity of cytokine interactions in retinal degeneration, combination therapies that target multiple inflammatory pathways may be more effective than monotherapies.

#### 4.4.1. Anti-VEGF + Cytokine Inhibitors

Anti-VEGF (e.g., aflibercept) + TNF-α inhibitors (e.g., adalimumab) may provide superior control of inflammation and vascular leakage in AMD and DR [16].

#### 4.4.2. Anti-Inflammatory Cytokines + Microglial Modulation

IL-10 therapy + CSF1R inhibitors have been suggested for sustained neuroprotection in RP and AMD [19].

#### 4.4.3. Personalized Cytokine Therapy

Advancements in cytokine profiling are enabling personalized treatments based on individual inflammatory markers, optimizing therapy for each patient [20].

### 4.5. Future Perspectives and Challenges

Despite advances in cytokine-targeted therapies, several challenges remain, including the following:Optimizing drug delivery: Intravitreal administration limits systemic side effects but requires repeated injections.Long-term safety: Suppressing inflammation may lead to increased infection risk or altered immune responses.Identifying patient subgroups: Not all patients respond equally to cytokine inhibition, highlighting the need for personalized medicine approaches.

Future research should focus on refining drug delivery methods (e.g., nanoparticles, sustained-release implants) and developing multi-targeted approaches to restore cytokine homeostasis and preserve vision in degenerative retinal diseases.

Targeting cytokines represents a promising strategy for retinal degenerative diseases, with therapies ranging from monoclonal antibodies (adalimumab, tocilizumab), gene therapy, and microglial modulation to combination approaches. While cytokine-targeted therapies are advancing rapidly, further research is needed to optimize safety, efficacy, and patient selection for these interventions [21,22,23,24].

Through continued innovation in precision medicine, sustained drug delivery, and genetic modulation, cytokine therapy may offer long-term disease modification and neuroprotection, improving outcomes for patients with AMD, DR, and RP.

Table 1 is a summarizing the therapeutic strategies discussed in the provided text for treating retinal degenerative diseases (AMD, DR, and RP). Note that some strategies are applicable to multiple diseases.

## 5. Conclusions

Degenerative retinal diseases, including age-related macular degeneration (AMD), diabetic retinopathy (DR), and retinitis pigmentosa (RP), are complex disorders driven by a combination of genetic, metabolic, and inflammatory factors. Among these, cytokine dysregulation has emerged as a critical component in disease pathogenesis, influencing inflammation, neurodegeneration, and vascular dysfunction.

The evidence reviewed in this work highlights the dual role of cytokines in retinal homeostasis and pathology. Pro-inflammatory cytokines, such as TNF-α, IL-1β, and IL-6, contribute to chronic neuroinflammation, retinal cell apoptosis, and vascular permeability, exacerbating disease progression. Conversely, anti-inflammatory cytokines, including IL-10, TGF-β, IL-4, and IL-13, serve as protective agents that counteract inflammatory damage and promote tissue repair. The imbalance between these cytokine networks determines disease severity, progression, and response to therapy.

Therapeutic advancements targeting cytokine pathways have shown promise in mitigating inflammation and preventing vision loss. Anti-VEGF therapies (e.g., ranibizumab, aflibercept, bevacizumab, and faricimab) have revolutionized the treatment of neovascular AMD and DR, reducing pathological angiogenesis and vascular leakage. Corticosteroids remain an essential tool for modulating inflammation and stabilizing the blood-retinal barrier. Furthermore, cytokine inhibitors, such as TNF-α blockers (adalimumab, infliximab), IL-6 antagonists (tocilizumab), and IL-1β inhibitors (canakinumab), are being actively explored for their potential to reduce chronic inflammation in retinal diseases.

Emerging strategies, including gene therapy, microglial modulation, and CRISPR/Cas9-based cytokine regulation, offer exciting prospects for personalized treatment. These approaches aim to restore cytokine homeostasis at a molecular level, potentially providing long-term disease modification and neuroprotection.

Despite these advancements, several challenges remain. The delivery of cytokine-targeted therapies to the retina without systemic side effects is a significant hurdle. Moreover, the individual variability in cytokine profiles necessitates a personalized medicine approach, integrating biomarker-guided therapies to optimize treatment efficacy.

In conclusion, understanding the role of cytokines in retinal degeneration has provided novel therapeutic insights and will continue to shape the development of next-generation therapies. Future research should focus on identifying precise cytokine targets, refining drug delivery systems, and integrating multi-modal treatment strategies to preserve vision and improve patient outcomes. Through continued advancements in cytokine biology and therapeutic innovation, the management of AMD, DR, and RP will evolve toward more effective, individualized, and long-lasting interventions.

## Figures and Tables

**Table 1 biomedicines-13-01724-t001:** Summary of therapeutic strategies targeting specific cytokines for the treatment of retinal degenerative diseases. The table outlines the mechanism of action and potential treatment examples for each strategy. Acronyms: AMD (Age-related Macular Degeneration), DR (Diabetic Retinopathy), RP (Retinitis Pigmentosa), TNF-α (Tumor Necrosis Factor-alpha), IL-6 (Interleukin-6), IL-1β (Interleukin-1 beta), CSF1R (Colony-Stimulating Factor 1 Receptor), VEGF (Vascular Endothelial Growth Factor) [17,18,29,30,31,32].

Therapeutic Strategy	Target Cytokine(s)	Disease(s)	Mechanism of Action	Potential Treatment Example
Cytokine Inhibitors			Directly blocks pro-inflammatory cytokines	
TNF-α Inhibitors	TNF-α	AMD, DR, RP	Neutralizes TNF-α, reducing inflammation	Adalimumab, Infliximab, Etanercept
IL-6 Inhibitors	IL-6	AMD, DR	Blocks IL-6, reducing inflammation and vascular leakage	Tocilizumab, Siltuximab
IL-1β Inhibitors	IL-1β	AMD, DR	Blocks IL-1β, reducing inflammasome activation and apoptosis	Canakinumab, Anakinra
Microglial/Immune Cell Modulation			Modulates microglial and immune cell activity to shift towards a neuroprotective phenotype	
CSF1R Inhibitors	Indirect (via microglia)	RP	Inhibits CSF1R, reducing microglial overactivation	Pexidartinib, PLX5622
Macrophage Polarization Therapy	Indirect (via macrophages)	AMD, RP	Promotes M2 macrophage polarization, reducing inflammation and promoting tissue repair	IL-4, IL-13
Gene Therapy	Varies	AMD, DR, RP	Alters cytokine expression at a genetic level	AAV-mediated IL-10, TGF-β gene therapy, CRISPR/Cas9-based TNF-α silencing
Anti-VEGF Therapy	VEGF	AMD, DR	Blocks VEGF, reducing neovascularization	Ranibizumab, Aflibercept, Bevacizumab, Faricimab
Corticosteroids	Multiple	DR	Reduces inflammation	Dexamethasone, Triamcinolone
Combination Therapies	Multiple	AMD, DR, RP	Combines multiple approaches for enhanced efficacy	Anti-VEGF + cytokine inhibitors, IL-10 therapy + CSF1R inhibitors

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
