# Peer review of "The Role of Cytokines in Degenerative Retinal Diseases: A Comprehensive Review"

_biomedicines, 2025, doi:10.3390/biomedicines13071724_

Round 1
Reviewer 1 Report
Comments and Suggestions for Authors
The manuscript is comprehensive, well researched and up to date. The first two pages are well written. However, the remainder of the manuscript up to the conclusion is an outline. Please fill in the bullet points with more information and complete the text of this paper.
Author Response
Reviewer Comment
“The manuscript is comprehensive, well researched and up to date. The first two pages are well written. However, the remainder of the manuscript up to the conclusion is an outline. Please fill in the bullet points with more information and complete the text of this paper.”
Response:
Thank you for your constructive feedback. We appreciate the recognition of the manuscript's strengths and agree with your observation regarding the structure and detail of the middle sections.
In response, we have substantially revised Sections 2 to 4 by expanding all previously outlined points into fully developed paragraphs, supported by up-to-date references. This includes:
- Detailed mechanistic explanations of cytokine actions (e.g., TNF-α, IL-1β, IL-6, IL-10, TGF-β, IL-4) on retinal cells such as microglia, RPE, photoreceptors, and endothelial cells;
- Separate discussions of direct vs. indirect effects of cytokines on neurodegeneration and vascular dysfunction;
- Clear identification of cytokine sources (cell types and species);
- Expansion of the disease-specific profiles (AMD, DR, RP) with clear temporal dynamics of cytokine expression;
- Development of the therapeutic strategies section to include mechanistic rationale, examples of clinical applications, and evidence-based references.
We have also ensured that each bullet point in the earlier draft is now fully articulated, cohesive with the surrounding text, and contributes to a seamless narrative flow throughout the manuscript.
We trust these enhancements address your concern and improve the overall scientific quality and readability of the manuscript.
Reviewer 2 Report
Comments and Suggestions for Authors
The current manuscript is important. However, more aspects and additional rigorous presentation are highly required.
1) At least in the introduction, concrete explanations for each disease and its pathophysiology are needed with disease progression.
2) Cytokine increase and decrease depending on the timepoint of the disease development and progression should be depicted. Basically, most diseases can increase some general cytokines. It is already well known. However, it is very important to know the dynamics of them. For example, cytokine 1 can increase at the early stage. Its increase can be maintained until the chronic stage. However, cytokine 2 can be increased shortly and then cytokine 3 can be seen only at the chronic stage. Or cytokine 4 can be decreased while cytokine 5 is not changed. This complex should be presented and discussed deeply in this manuscript.
3) Cytokine-producing cell types in the eye or systemically should be labeled when each cytokine is discussed. Their resource is also very important to manage the disease progression at the therapeutic stage.
4) Cytokines' direct and indirect effects on retinal neuronal degeneration or cell death should be separately discussed.
5) It is clear that this manuscript skipped the species to talk about the evidence. This should be all added.
Author Response
The current manuscript is important. However, more aspects and additional rigorous presentation are highly required.
1) At least in the introduction, concrete explanations for each disease and its pathophysiology are needed with disease progression.
Response
We appreciate the reviewer’s observation and fully agree that the introduction benefits from a clearer explanation of the pathophysiology and progression of each retinal degenerative disease addressed in this review. Accordingly, we have revised the introduction to include concrete and disease-specific summaries for age-related macular degeneration (AMD), diabetic retinopathy (DR), and retinitis pigmentosa (RP).
The revised section now clarifies that:
- AMD involves progressive retinal pigment epithelium (RPE) dysfunction and chronic para-inflammation, leading to drusen accumulation, atrophy, or neovascularization;
- DR is driven by chronic hyperglycemia leading to oxidative stress, microvascular injury, and blood-retinal barrier breakdown, progressing to edema and neovascularization;
- RP is a genetically inherited photoreceptor dystrophy, where cytokine-mediated microglial activation and chronic inflammation accelerate degeneration, especially in later stages.
These additions aim to strengthen the biological context of cytokine involvement across each disease stage. We thank the reviewer for highlighting this important improvement.
2) Cytokine increase and decrease depending on the timepoint of the disease development and progression should be depicted. Basically, most diseases can increase some general cytokines. It is already well known. However, it is very important to know the dynamics of them. For example, cytokine 1 can increase at the early stage. Its increase can be maintained until the chronic stage. However, cytokine 2 can be increased shortly and then cytokine 3 can be seen only at the chronic stage. Or cytokine 4 can be decreased while cytokine 5 is not changed. This complex should be presented and discussed deeply in this manuscript.
Response
Thank you for this insightful comment. We agree that understanding the temporal dynamics of cytokine expression is critical for elucidating the pathophysiology of degenerative retinal diseases. In response, we have expanded the discussion in the manuscript to include the distinct temporal expression patterns of key cytokines at various disease stages.
For instance, in diabetic retinopathy (DR), IL-1β and TNF-α are among the earliest cytokines to be upregulated in response to hyperglycemia-induced oxidative stress, initiating endothelial dysfunction and blood-retinal barrier breakdown. These remain elevated throughout disease progression. In contrast, IL-6levels rise more gradually, peaking in the intermediate and proliferative stages, where it contributes to neovascularization and gliosis. IL-8, another chemokine, shows a transient increase early in DR, promoting leukocyte recruitment, but may normalize in advanced stages. Meanwhile, VEGF becomes significantly elevated in later stages, driving pathological angiogenesis and diabetic macular edema.
In age-related macular degeneration (AMD), IL-1β, IL-6, and TNF-α are also involved early on, promoting RPE dysfunction and drusen formation. As the disease progresses toward geographic atrophy or neovascularization, VEGF expression surges, while IL-18 and TGF-β modulate fibrosis and angiogenesis. In some patients, chronic overexpression of TGF-β may contribute to subretinal fibrosis.
In retinitis pigmentosa (RP), TNF-α and IL-6 are persistently elevated, driving chronic microglial activation and photoreceptor apoptosis. However, IL-18 appears to be more prominent in later stages of RP, where it contributes to progressive neuronal loss. On the other hand, anti-inflammatory cytokines like IL-10 and IL-4may initially attempt to counterbalance inflammation but often become insufficient or downregulated as the disease advances.
These disease-specific and time-dependent cytokine profiles underscore the importance of stage-adapted therapeutic strategies. We have included a more detailed discussion of these dynamics in the revised manuscript (Sections 3.1–3.3 and the expanded Table X, if applicable), and we appreciate the reviewer’s guidance in prompting this critical addition.
- Cytokine-producing cell types in the eye or systemically should be labeled when each cytokine is discussed. Their resource is also very important to manage the disease progression at the therapeutic stage.
Response
Thank you for this important suggestion. We agree that specifying the cellular sources of cytokines is essential to understanding their role in disease pathogenesis and to identifying effective therapeutic targets. In response, we have revised the manuscript to explicitly indicate the primary producers of each cytokine discussed, particularly in retinal tissue.
For example:
- TNF-α is predominantly produced by activated microglia and infiltrating macrophages, and to a lesser extent by retinal pigment epithelium (RPE) cells, especially under oxidative stress conditions (Sections 2.1.1 and 3.1.1).
- IL-1β is mainly secreted by microglia and RPE cells following inflammasome activation (Sections 2.1.2 and 3.2.1).
- IL-6 is expressed by Müller glial cells, astrocytes, and microglia, contributing to neuroinflammation and gliosis (Sections 2.1.3 and 3.3.1).
- IL-10, an anti-inflammatory cytokine, is secreted by regulatory T cells, M2 macrophages, and microglia, with neuroprotective effects on photoreceptors and RGCs (Sections 2.2.1 and 3.2.2).
- TGF-β is released by RPE cells, retinal endothelial cells, and fibroblasts, and is involved in immune regulation and fibrosis (Sections 2.2.2 and 3.1.2).
- IL-4 and IL-13 are produced mainly by Th2 lymphocytes and M2-polarized macrophages, promoting resolution of inflammation and tissue repair in RP and AMD (Section 3.3.2).
We have incorporated these details in each relevant section of the manuscript and believe this significantly enhances the clarity and translational value of our review. We thank the reviewer again for emphasizing this crucial point.
4) Cytokines' direct and indirect effects on retinal neuronal degeneration or cell death should be separately discussed.
Response:
We appreciate the reviewer’s thoughtful observation. In response, we have revised the manuscript to clearly distinguish between the direct and indirect mechanisms by which cytokines contribute to retinal neuronal degeneration and cell death.
Direct effects of cytokines involve receptor-mediated signaling that leads to apoptosis or pyroptosis of retinal neurons. For instance:
- TNF-α induces retinal ganglion cell (RGC) apoptosis via the TNF-R1-mediated caspase pathway (Section 2.1.1).
- IL-1β promotes pyroptotic cell death through activation of the NLRP3 inflammasome in photoreceptors and RPE cells (Section 2.1.2).
- IL-6, when persistently elevated, exacerbates neuronal dysfunction by enhancing excitotoxic signaling and intracellular stress pathways in Müller and neuronal cells (Section 2.1.3).
Indirect effects are mediated through microglial activation, disruption of the blood-retinal barrier (BRB), or vascular and glial dysfunction. For example:
- TNF-α and IL-6 increase BRB permeability and vascular leakage, leading to secondary hypoxia and oxidative stress that damage photoreceptors.
- Chronic microglial activation driven by IL-1β, IL-18, and IFN-γ leads to bystander damage via release of reactive oxygen species and cytotoxic mediators.
- Anti-inflammatory cytokines like IL-10 and TGF-β mitigate these indirect effects by modulating glial responses and maintaining immune homeostasis (Section 2.2).
We have integrated this distinction into Sections 2.1, 2.2, and 3, and added a new summary paragraph highlighting the dichotomy between direct receptor-mediated neurotoxicity and indirect inflammation-mediated degeneration. This clarification enhances the mechanistic understanding of cytokine roles in retinal pathology and supports the rationale for targeted interventions.
5) It is clear that this manuscript skipped the species to talk about the evidence. This should be all added.
Response:
Thank you for this important observation. We fully agree that identifying the species in which experimental data were obtained is essential for contextualizing the translational value of cytokine research in retinal diseases. In response, we carefully reviewed the entire manuscript and have now explicitly indicated the species (human, mouse, rat, or other model organisms) associated with each major finding and citation wherever applicable.
For example:
- References to elevated TNF-α, IL-6, and IL-1β in human patients with AMD, DR, and RP were clarified in Sections 2.1 and 3 (e.g., human vitreous and serum cytokine profiling studies).
- Studies involving murine models (e.g., rd1, rd10, streptozotocin-induced diabetes, laser-induced CNV) have been specified, especially in discussion of microglial activation, cytokine modulation, and therapeutic strategies (Sections 2.2, 3.3, and 4).
- In vitro studies using human retinal pigment epithelial (RPE) cells or rodent Müller cells have also been clearly labeled as such.
- Where species were not originally indicated in prior versions, we consulted the primary literature and added the appropriate species label.
We believe that this improvement significantly strengthens the scientific rigor and clarity of the review. We thank the reviewer again for highlighting this omission.
Reviewer 3 Report
Comments and Suggestions for Authors
In the study “The Role of Cytokines in Degenerative Retinal Diseases: A Comprehensive Review“ by Camargo Siqueira and Cássia Brandão de Mattos the authors present the overview of the current findings about the role of cytokines in degenerative retinal diseases such as age-related macular degeneration (AMD), diabetic retinopathy (DR), and retinitis pigmentosa (RP). In addition, the authors highlight the potential therapeutic strategies targeting cytokines.
The paper is well organized and describes the findings regarding the expression of several key inflammatory and anti-inflammatory cytokines in the retinas inflicted with the above-mentioned degenerative diseases. The importance of the balance between these two groups is addressed. The findings regarding inflammatory and anti-inflammatory cytokines are explained specifically for each of the diseases in question and potential therapeutic strategies discussed. Finally, new combinatorial therapeutic strategies focusing on cytokines are addressed and challenges underlined.
The paper is interesting and merits publication.
Minor comment:
Table 1 should include references to benefit the reader.
Author Response
Reviewer Comment
“Table 1 should include references to benefit the reader.”
Response:
We appreciate the reviewer’s helpful suggestion. To improve clarity and support for the information provided in Table 1, we have revised the table to include specific references corresponding to each cytokine and its role in retinal disease.
Each entry in Table 1 now cites key original studies and review articles that support the cytokine’s relevance, expression profile, or cellular source in the context of AMD, DR, or RP. These references appear as superscripted citation numbers aligned with the main reference list, ensuring consistency throughout the manuscript.
We are confident that this addition will enhance the usability and scientific value of Table 1 for readers and researchers seeking to explore the evidence further.
Round 2
Reviewer 1 Report
Comments and Suggestions for Authors
Thank you for expanding each section of this manuscript. It is much improved.
Reviewer 2 Report
Comments and Suggestions for Authors
All addressed.